# Mind over Microplastics: Exploring Microplastic-Induced Gut Disruption and Gut-Brain-Axis Consequences

Charlotte E. Sofield [1] , Ryan S. Anderton [1,2] and Anastazja M. Gorecki [1,*]

[1] School of Health Sciences, University of Notre Dame Australia, Fremantle, WA 6160, Australia; charlotte.sofield@nd.edu.au (C.E.S.); ryan.anderton@nd.edu.au (R.S.A.)
[2] Institute for Health Research, University of Notre Dame Australia, Fremantle, WA 6160, Australia
* Correspondence: anastazja.gorecki@nd.edu.au

**Abstract:** As environmental plastic waste degrades, it creates an abundance of diverse microplastic particles. Consequently, microplastics contaminate drinking water and many staple food products, meaning the oral ingestion of microplastics is an important exposure route for the human population. Microplastics have long been considered inert, however their ability to promote microbial dysbiosis as well as gut inflammation and dysfunction suggests they are more noxious than first thought. More alarmingly, there is evidence for microplastics permeating from the gut throughout the body, with adverse effects on the immune and nervous systems. Coupled with the now-accepted role of the gut-brain axis in neurodegeneration, these findings support the hypothesis that this ubiquitous environmental pollutant is contributing to the rising incidence of neurodegenerative diseases, like Alzheimer's disease and Parkinson's disease. This comprehensive narrative review explores the consequences of oral microplastic exposure on the gut-brain-axis by considering current evidence for gastrointestinal uptake and disruption, immune activation, translocation throughout the body, and neurological effects. As microplastics are now a permanent feature of the global environment, understanding their effects on the gut, brain, and whole body will facilitate critical further research and inform policy changes aimed at reducing any adverse consequences.

**Keywords:** microplastics; gut; brain; microbiome; neurodegeneration



## 1. Introduction

Plastic has revolutionized life since the 1950's—commencing with the widespread implementation of polystyrene, polyester, and polyvinyl chloride—and its durability and versatility have since been unmatched. Despite the longevity of plastic, it is the primary material used for single use and disposable items, leading to an enormous amount of plastic waste [1]. By 2050, plastic production is projected to reach 1.1 billion tonnes per year, with an expected accumulation of 12 billion tonnes of plastic waste in landfills. Large pieces of plastic (primary plastics) are weathered and degraded to form millions of secondary microplastics (diameter < 1 mm) which vary in size, shape, and texture. Such microplastics have been found throughout the environment, including deep sea trenches, high altitude mountain water systems, and even beehives [2–4]. Environmental pollution leads to the ubiquitous presence of microplastics in food and drinking water, meaning oral ingestion is a major exposure route for the human population. Alarmingly, environmental surveys estimate that humans ingest up to 5 g of microplastics a week [5], however, research on the effects of this exposure in humans is still in its infancy [6,7]. A growing number of studies suggest that oral microplastic exposure leads to gut microbiome dysbiosis, gastrointestinal absorption, immune activation, and deposition in various tissues, including the brain, with neuronal dysfunction and damage [8–12]. These troubling observations support the hypothesis that population-wide chronic microplastic exposure, and consequent systemic inflammation, is a contributing factor to the increasing rates of neurodegenerative disease [13]. This comprehensive narrative review will explore current knowledge concerning

how oral microplastics affect the gut-brain-axis, considering evidence from the gut lumen to the brain, in various models, tissues, and conditions.

## 2. Introducing the Gut-Brain-Axis

The gut (directly) and gut microbiome (indirectly) communicate bi-directionally with the nervous system, forming the gut-brain-axis (GBA). Together with the gut microbiota, the gut is vital in whole-body health as it facilitates nutrition, regulates the immune system, and produces critical hormones that influence various bodily systems. Importantly, lumen contents are separated from the host's internal biology by a single layer of absorptive epithelial cells [14,15]. This dynamic and selectively semi-permeable gut barrier is regulated by the gut environment, evidenced by modifiable expression of tight junction proteins such as claudins (claudin 1, claudin-2, claudin-15), occludin, zonular occludens-1, and junction adhesion molecules, leading to changes in intestinal permeability [16]. Due to the close contact with pathogens in the lumen, the gut houses a large population of immune cells, which are differentially activated by pathogens, bacterial toxins, and commensal organisms [17,18]. The gut communicates with the brain via the vagus nerve, and through the systemic circulation of metabolites, hormones, and immune factors—forming the GBA. Under 'normal' conditions, the brain is protected from circulating toxins, pathogens, and inappropriate immune activity by the blood-brain-barrier (BBB), another selectively semi-permeable network of endothelial cells linked by tight junctions [19]. Importantly, metabolites from gut probiotics protect and support the BBB [20,21], bolstering the connection between the gut and the brain.

## 3. Mechanisms of Microplastic Damage to the Gut

Despite the complexity and resilience of the gut, various environmental factors and stressors can cause dysfunction of the gut and immune system. In turn, the dynamic GBA can be perturbed; for example, chronic systemic inflammation originating in the gut, or a 'leaky gut condition', increases permeability of the BBB, initiating harmful cellular and protein changes in the brain [19,22,23]. Such gut-driven inflammation is implicated in the pathogenesis of neurodegenerative disease including Alzheimer's disease (AD) and Parkinson's disease (PD), where exposure to environmental pollutants (e.g., pesticides) is a crucial trigger in this pro-inflammatory cycle [18,23–25]—although the influence of orally ingested microplastics has not been extensively investigated. Chronically ingested by humans through contaminated food and beverages [26], microplastics are a possible, and ever-present, disruptor of the gut environment. Importantly, a homeostatic gut microbiome can be perturbed by changes to the luminal environment [27], meaning microplastics pose a substantial threat to microbial ecology in the gut. Similarly, the enteric immune system is activated by intestinal damage and dysbiosis which can result from oral microplastic exposure [28]. As such, studies investigating oral microplastic exposure have indicated that dietary microplastics interact with the gut via tissue accumulation, microplastic-induced toxicity, immune activation, histological damage, and microbiome changes, discussed below. Microplastic-induced gut changes can affect the brain and body [29–31]; but is it not known how much damage is directly caused by microplastic accumulation within cells and tissues, compared to functional changes originating in the gut [32–34] and GBA disruption.

### 3.1. The Gut Microbiome

The gut microbiome is a critical mediator of health, and thus an important modulating factor of disease associated with the GBA. For example, gut dysbiosis has been widely correlated with the pathogenesis of IBD, as well as mental health and neurodegenerative conditions [24,35]. Environmental factors can contribute to gut dysbiosis, and new evidence shows that microplastics are a selective factor among bacterial communities. For example, microbiological studies have shown that microplastics promote the expansion of specific species, particularly pathogens and opportunistic pathogens [36,37]. Microplastics also facilitate the formation of unique microbial biofilms, further demonstrating that microbe-

microplastic interactions are species-specific [36]. Due to this direct effect on bacteria, growing research has interrogated microplastic-induced community alterations to the gut microbiome.

Evidence now shows that microplastics significantly disturb the gut microbiome. The four dominant phyla of the human intestinal microbiome are *Firmicutes*, *Bacteroidetes*, *Actinobacteria*, and *Proteobacteria* [38]. In mice and zebrafish, all four key phyla are repeatedly affected by gastrointestinal microplastic exposure across several studies (Figure 1); the relative abundance of *Bacteroidetes* is consistently reduced by microplastic exposure, however, trends for the other three key phyla are more variable. Importantly, the *Bacteroidetes* phylum has an anti-inflammatory effect on the gut, and is also reduced in IBD and cystic fibrosis [39,40]. Most studies show that microplastic exposure reduced species richness and diversity [41–45], whereas some report that a higher dose of microplastics improved species richness [46,47]. Greater microbial diversity confers stability and health in the gut, whereas a loss of diversity is often a characteristic of an unhealthy gut state associated with IBD and recurrent *Clostridium difficile* infections [27]. Mice exposed to polyethylene microplastics for three weeks showed greater intra-group species differences than the control, in these mice the highest microplastic dose caused the greatest change in diversity [46]. This means that despite a perceived improvement to the structure of the microbiome in some cases, overall, microplastic exposure de-stabilises microbial ecology in the intestine.

Additionally, the microplastic shape (fragmented or spherical) modulates microplastics-microbiome interactions. Using zebrafish, Guo et al. [42] found that spherical microplastics caused greater microbial disruption than fragmented microplastics; whereas Qiao et al. [48] reported the opposite effect despite using the same model, with comparable doses (1 μg/L–1 mg/L) and exposure periods (four and three weeks, respectively). Interestingly, the findings from a computer-controlled dynamic in vitro model of gastrointestinal digestion, showed that microbes were differentially affected by microplastics between segments of the colon [49]. Additionally, a quasi-experimental study with human participants (*n* = 30) found that significant microbiome changes at a phylum level (compared to the control group; *n* = 30), only occurred one month after a period of high microplastic exposure ceased [50]. Faecal microplastic concentration more than halved between the high exposure period and cessation, however, one month later microbial diversity was significantly reduced, as was the abundance of key phyla: *Firmicutes* and *Bacteroidetes* [50]. This suggests that microplastics retained in the gut continue to exert effects long after their initial ingestion, and that microbiome composition takes time to re-stabilise [50]. Such delayed aftereffects of microplastic consumption have not been well-studied, as animals are usually sacrificed immediately at the end of the exposure period—an important consideration for future research. Despite differences in trends for community indexes and taxonomic groups, overall, there is consistent evidence demonstrating that microplastic exposure significantly changes the structure of the gut microbiome.

The Gut Microbiome: Functional Effects

When the stability of the gut microbiome is disrupted, metabolic, immune, and nervous systems are impaired. In a range of studies, microplastic-induced gut dysbiosis modified differentially expressed genes and metabolites which translated to altered Kyoto Encyclopedia of Genes and Genome (KEGG) pathways involved in lipid, nucleic acid, and hormone metabolism, protein secretion, neurotoxicity, inflammation, aging, metabolic disease, and cancer [41–44,48,50]. Increased intestinal permeability is another functional change associated with microbial dysbiosis secondary to microplastic exposure [33,44,45,51,52]. In two different mouse models of peripheral disease (vascular calcification and kidney disease), both microplastic-induced intestinal permeabilisation and dysbiosis caused higher serum levels of lipopolysaccharide (LPS)—a pathogen associated molecular pattern (PAMP) derived from enteric bacteria which drives systemic inflammation [33,53]. Interestingly, antibiotics and faecal microbiota transplant in mice reversed microplastic-induced gut barrier damage and microbiome disruption, respectively [33,52],

demonstrating that dysbiosis underpins microplastic-induced gut dysfunction and inflammation. Mounting evidence has associated microplastic ingestion with dysbiosis and intestinal damage, which translates to an array of functional changes in the GBA and the rest of the body.

### 3.2. Uptake of Microplastics from the Gut Lumen

A healthy intestinal barrier should prevent the passage of microbes and foreign material from the gut lumen into circulation. However, studies using fluorescent microplastics in mice and zebrafish demonstrate the transit of microplastics from the gut lumen, resulting in their accumulation in intestinal cells and tissues in a size- and shape-dependent manner [34,48,54–56]. Notably, smaller microplastics (<5 μm) were taken up by the intestinal epithelium at a greater rate than larger microplastics (≥5 μm) [56–59]. This is important clinically, as although the size of environmental microplastics varies considerably, the majority (up to 95%) of microplastics in drinking water are smaller than 10 μm [60–62]. Studies investigating the effect of microplastic shape have found that compared to pristine spherical microplastics, microplastics with sharp, irregular edges caused more severe membrane damage and accumulated at a higher rate [63]. Importantly, microplastic contaminants in food and drinking water have been weathered and fragmented into irregular fragments and fibres, and altered further by gastrointestinal digestive processes [64,65]. Consequently, the use of spherical microplastics in the majority of current research may not be clinically relevant [64,65] and could be underestimating true tissue accumulation in humans.

### 3.2.1. Mechanism of Microplastic Uptake

The mechanism by which microplastics are transported from the gut lumen into tissue and around the body is not clear. Intestinal transwell models have reported that microplastics interact with epithelial villi and cross the monolayer without being internalized [66,67], indicating that microplastics navigate the intestinal barrier via paracellular transport. Absorption is one of the gut's primary functions, therefore microplastic internalisation by intestinal epithelial cells has been investigated as another possible route of uptake [56,59,67,68]. Confocal microscopy is a favoured technique to assess microplastic internalisation in vitro as it provides a three-dimensional view of the microplastics' position within the cell. This imaging has shown that microplastics readily adhere to the surface of cells in vitro, and that microplastics up to 10 μm can be internalised by intestinal cells after only 12 h of exposure [56,58,69]. Some evidence indicates that a smaller range microplastics can be transported through the cell in lysosomes [59], although this has not yet been replicated. Similarly, the inhibition of the ATP-binding cassette (ABC) transporter in Caco-2 cells increased the intracellular accumulation of 0.1 μm microplastics, indicating that this efflux pump may assist with removing small microplastics from cells [59]. However, based on the fractional quantities of microplastics that are internalized in these studies, it is unlikely that intracellular transport is the main route by which microplastics move from the gut lumen to peripheral circulation. Rather, pre-existing gut barrier breakdown (induced by diet, microplastics, and more) may facilitate uptake via paracellular pathways, as previously discussed. Hence, in vitro models of microplastic transport are limited because they cannot replicate the complexity of the dynamic gut environment which ultimately dictates intestinal permeability.

### 3.2.2. Environmental and Luminal Factors Influence Microplastic Accumulation in the Gut

In vitro studies also fail to recapitulate how contact with environmental and gastrointestinal factors modifies microplastic effect. For example, microplastics interact with factors found within the environment and gut lumen, with studies reporting the formation of coronas (e.g., Bio-corona, eco-corona) on the surface of microplastics comprising salts, proteins, lipids, heavy metals, antibiotics, microbes, and other molecules present in the environment and biological fluids [69–73]. The effect of such coronas on gastrointestinal uptake is unknown, but it is hypothesized that this is a mechanism for microplastics to

act as 'carriers' of toxins from the environment into the gastrointestinal system [74,75]. Additionally, a mouse study found that *Helicobacter pylori* forms a biofilm on the surface of digested polyethylene microplastic fragments, which synergistically enhanced both tissue accumulation of microplastics and *H. pylori* colonisation of the gastric and intestinal epithelium [76]. Interestingly, coupling four weeks of high fat diet with oral microplastic exposure in mice exacerbated microplastic accumulation in the intestinal wall, intestinal permeability, thinning of the mucin layer, and intestinal inflammation, compared to microplastic-exposed mice fed a normal diet [68]. Diet is an important regulator of gastrointestinal function in the gut, and may protect or predispose one to microplastic-induced inflammation and dysbiosis [68,77]. Thus, this research supports the hypothesis that microplastic absorption of surrounding matter affects the way microplastics behave throughout the body, but greater inquiry into the composition and mechanisms of microplastic-coronas in the gut, brain, and peripheral circulation is still necessary.

### 3.3. Microplastic Toxicity

Using a variety of sizes, shapes, and exposure periods, studies have concluded that microplastics do not cause death of intestinal cells and tissues [56,58,59,63,78–80]. Although intestinal cell death was evident after exposure to high microplastic doses in vitro [56,63,79,80], these concentrations were not environmentally relevant. This research suggests that dietary microplastics influence the gut in ways other than inducing cell death. For example, microplastic exposure raised levels of reactive oxygen species (ROS), increased superoxide dismutase (SOD) activity, and upregulated the expression of genes involved in oxidative stress in the gut [48,58,59,80]. Although microplastics are not lethal to cells of the gut epithelium, the cellular stress that microplastic exposure induces has structural and functional consequences that threaten gut health overall.

### 3.4. Histological Alterations in the Gut

The presence of microplastics in the gut lumen causes structural changes to the intestinal epithelium and mucosa (Figure 1). Histological analysis has demonstrated microplastic-induced inflammation characterised by oedema, vacuolisation, loose tissue glands, increased crypt depth, villi cracking, enterocyte splitting, cilia defects, and small vessel proliferation from the duodenum to the colon [44,47,48,57,65]. However, some research reports that only fragmented or fibrous microplastics induced significant histological changes to the gut epithelium, with no effect from spherical microplastics [48,57,81]. When investigating affected cell populations, six weeks of microplastic exposure in mice downregulated absorptive epithelial cell and enteroendocrine cell markers [65], which would impair nutrient absorption and gut hormone production, respectively. Microplastic exposure also altered the abundance of mucus-producing cells and mucus volume throughout the intestine [48,65,68,82]. While some studies showed that microplastic exposure resulted in lower *Mucin 2* expression (*MUC2*; a gene encoding a marker of mucin secreting cells), goblet cell coverage, and mucous volume [48,68], others found that transcript levels of mucin-2 protein and mucous volume increased following microplastic exposure [65,82]. These differing results may be due to the use of different doses, sizes, and shapes, as discussed above. Nonetheless, controlled mucous secretion is vital for enteric immune function and maintenance of the gut barrier. Deterioration of intestinal cell populations and interrupted mucous secretion impairs the gut's ability to regulate what nutrients, toxins, and other molecules pass from the gut lumen into systemic circulation. Interestingly, this microplastic-induced 'leaky gut' condition is also associated with peripheral and neuro-inflammation in diseases like PD and AD through disruption of the BBB.

**Figure 1.** *Representative schematic summarizing the intestinal and peripheral effects of oral microplastic exposure.* (**A**) Pre-clinical evidence indicates that in the gut lumen, microplastics have been shown to have three major effects: (1) microbial dysbiosis, characterized by poorer diversity and compositional instability [41–45], and (2) increased gut permeability due to intestinal cell damage (namely enterocytes) [44,47,48,57,65]. These microplastic-induced disruptions to the microbiome and gut barrier contribute to gastrointestinal inflammation (3), leading to the translocation of pro-inflammatory cytokines, bacterial toxins (lipopolysaccharide, LPS), and mucosal immune cells from the gut into peripheral circulation microplastics [31,33,56,63]. After oral microplastic exposure, studies also report peripheral effects: microplastic circulation, microplastic deposition in peripheral organs, elevated serum LPS, and systemic inflammation [33,45]. (**B**) Notably, clinical evidence using human biopsy samples and post-mortem samples, microplastics have been isolated from the heart, saphenous vein, liver, spleen, kidney, gut, lower limb joints, and reproductive organs (represented by green zones) [8,10,53,83–88]. Figure created with Biorender.com.

### 3.5. Enteric Immune Activation

The enteric immune system is the first line of defence against pathogens and toxins that pass through the gastrointestinal lumen. Innate immune activation in the gut generates pro-inflammatory cytokines which drive local gut inflammation and also exert systemic effects through the circulatory system. Therefore, the immune response to microplastics in the gut is key to understanding their impact on whole-body health and the GBA. In mouse and zebrafish models of oral microplastic exposure, structural changes to the gut are accompanied by the infiltration of lymphocytes (T cells and natural killer cells), plasma cells, and mast cells into the intestinal mucosa, while anti-inflammatory mucosal macrophages become less abundant (Figure 1) [47,48,65,68]. Consequently, microplastic exposure promotes the production of pro-inflammatory cytokines including Tumour Necrosis Factor (TNF), Interferon gamma (IFN-$\gamma$), Interleukin 6 (IL-6), and Interleukin-1 beta (IL-1$\beta$) both in vitro and in vivo [59,65,68]. Similarly, the upregulation of innate immune receptor proteins Toll-like receptor 4 (TLR-4) and interferon regulatory factor 5 (IRF5) is evident after extended microplastic exposure (5+ weeks) in the mouse gut [47,76]. This activation of inflammatory immune cells, pro-inflammatory chemicals, and receptors indicates that gastrointestinal microplastic exposure activates the innate immune system. These findings are supported by clinical evidence showing a positive correlation between faecal microplastic concentration and IBD status and severity [12]. This intriguing observation highlights the possibility that dietary microplastics drive disease in the gut which possibly leads to inflammation and dysfunction around the body.

## 4. Microplastics Travel away from the Gut to Induce Peripheral Disease

### 4.1. Microplastic Disperse around the Body

New research isolating microplastics from human tissue throughout the body has recently been documented and widely discussed. Clinical evidence has found that microplastics in the blood are pumped through the circulatory system [88], and are deposited in the heart and blood vessels [53,86–88]. The circulation of microplastics also leads to their accumulation in peripheral sites including the liver, kidneys, spleen, joints, and reproductive organs (Figure 1) [8,10,83–85]. Furthermore, the discovery of microplastic particles in human placenta and amniotic fluid is particularly alarming, as the effect of microplastics on human embryonic or foetal development is unknown [10,83,84]. The microplastic fragments extracted from these human tissues were diverse in size, polymer type, and density. For example, microplastics found in the lower limb joints were relatively large ($\approx$50 μm on average) with a mean abundance of $\approx$5 particles/g of tissue [85], while urine samples had a far smaller size range of 4–15 μm, likely due to glomerular filtration [89]. These clinical studies identified ingestion as the source of microplastics in peripheral tissues, having accounted for background contamination during surgery and sample collection.

Microplastic contamination of human tissues has been well-replicated in mouse models, where fluorescent microplastics were swallowed and entered the gastrointestinal tract, before crossing the gut barrier and dispersing peripherally [31,32,34,54,76,90–93]. After only seven days (and up to a month) of exposure, these fluorescent microplastics were subsequently detected in the brain, liver, lungs, heart, kidney, and spleen of exposed mice [78,90,92]. However, microplastic longevity in tissues is not well-understood. One study found that for orally administered microplastic particles of 5 μm and 20 μm diameters, levels plateaued in various tissues after 14 days of exposure, remained stable until cessation at day 30, and were still present one week after exposure ceased [34]. In the kidney and gut, small (5 μm diameter) microplastics accumulated the most; whereas in the liver, large microplastics were more concentrated, reiterating that the biological accumulation of microplastics varies between organs and tissues. The alarming takeaway of clinical and in vivo research is that orally ingested microplastics enter most, if not all body systems, where they exert unique effects and drive dysfunction.

*4.2. Microplastics Induce Peripheral Disease*

The potential for microplastic-induced disease is multifaceted. Firstly, microplastics within peripheral tissues interact with and enter cells, where they induce primary cell injury, functional changes, and immune activation. Peripheral macrophages in the blood readily engulf microplastics whilst peripheral blood mononuclear cells (PBMC) release pro-inflammatory cytokines (e.g., IL-6 and TNF-$\alpha$), especially in response to fragmented microplastics [56,63]. A study using both spontaneous lupus-prone (MRL/lpr) and healthy mice found that the inflammatory response to microplastic exposure was accompanied by elevated serum autoantibodies indicative of systemic lupus erythematosus in both groups [94]. Microplastics also impair the cardiovascular system, as faecal microplastic concentration was positively correlated with vascular calcification score (in 47 patients with and without vascular calcification), while in rats, microplastic exposure aggravated existing vascular calcification and induced new mild vascular calcification [45]. Oral microplastic exposure in mice resulted in pathological kidney and liver histology and altered biomarkers of chronic renal disease and liver damage [31,33]. Additionally, mechanistic, in vitro studies observed inflammation, apoptosis, and cytotoxicity in liver and kidney cells after microplastic exposure [95,96]. In rodents, disease was associated with disruption of the gut-liver-axis and gut-kidney-axis via microplastic-induced intestinal dysbiosis, and consequent changes to key gut metabolites and pathways associated with gut-liver and gut-kidney function [31–33,45,94,97]. Metabolic balance is critical to maintain healthy systems, therefore metabolic alterations are another mechanism by which microplastics exacerbate peripheral disease. Moreover, the consequent elevation of LPS and pro-inflammatory cytokine levels in the blood exposes peripheral organs, and the BBB, to potent inducers of inflammation. Prolonged exposure to microplastic-induced peripheral inflammation, LPS leakage from the gut, and altered metabolites in circulation disrupts the integrity of the BBB, creating a hazardous environment for the brain.

## 5. The Neurotoxic and Neurodegenerative Effects of Oral Microplastic Exposure

*5.1. Microplastics cross the BBB*

Microplastics circulating in the blood inevitably encounter the BBB, which is susceptible to disruption by environmental pollutants directly and associated chronic inflammation (Figure 2). Shan et al. [93] found that microplastic exposure in an in vitro BBB model downregulated the expression of tight junction proteins zonulin and occludin. This has been reflected in vivo where microplastic exposure increased permeability of the BBB in the hippocampus, hypothalamus, and cortex of mice [78,91,93]. Microplastics of up to 10 μm enter the cortex, hippocampus, and cerebellum of mice in a dose-dependent manner after only 24 h of exposure [78,90,91]. Although this evidence was primarily derived from animal models, microplastic infiltration into brain tissue has also been observed in situ. Microplastics found in the brains of wild estuarine Seabass were mostly fragments smaller than 50 μm, although the largest particle found was 96 μm [98]. No data has been collected to show whether microplastics accumulate in the brains of humans although their presence in the placenta and urine (having crossed endothelial barriers) [10,83,99] indicates this possibility. Brain tissue is highly sensitive to foreign material, so microplastic transport across the BBB poses a significant risk for neuroinflammation and cellular changes in the brain.

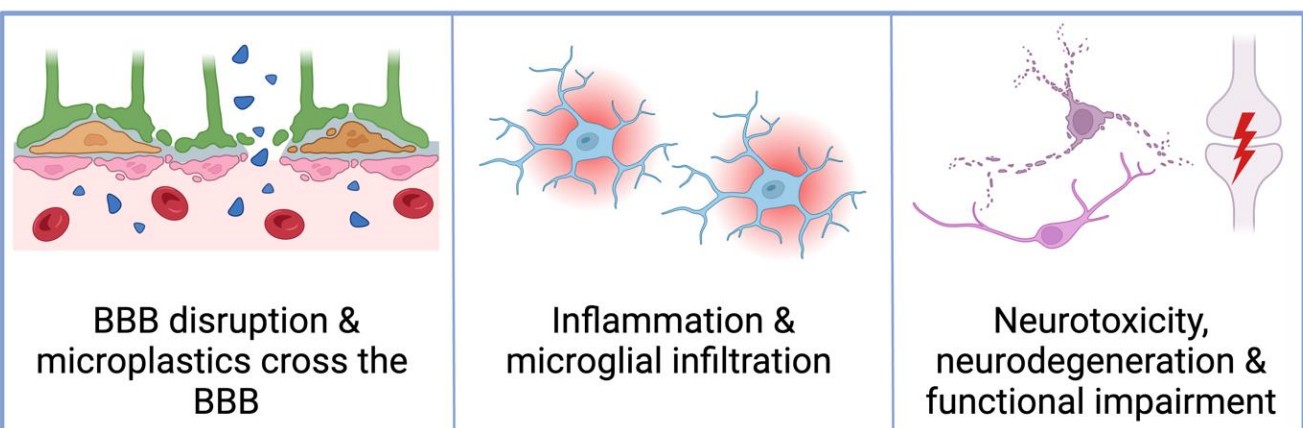

**Figure 2.** *Schematic representing microplastic induced neuroinflammation*. The blood-brain-barrier comprises microvascular endothelial cells (pink), astrocytes (green), and pericytes (orange), which shields the brain from toxins in the blood and unnecessary damage from the peripheral immune system [19]. Microplastic-induced blood-brain-barrier (BBB) disruption occurs via direct exposure to microplastics in the blood [88], or persistent peripheral inflammation resulting from dysbiosis and gut barrier deterioration [78,91,93,100]. After BBB disruption, microplastics contribute to inflammation, microglial activation [92,93], neurotoxicity [90,92,93,101,102], and neurodegeneration [103–105], with a functional impairment of neurons and synapses also reported. Figure created with Biorender.com.

### 5.2. Neuroimmune Activation

Around the body, microplastic tissue accumulation activates inflammation and attracts immune cell infiltration—it appears the brain responds in a similar way. Microglia are the macrophages of the brain and the first responders to foreign bodies that may have evaded the BBB [106]. It has been observed that microglia phagocytose microplastics and induce inflammation when microplastics are present in brain tissue (Figure 2) [92,93]. Oral microplastic exposure caused microglia to become activated and polarised, characterised by a hypertrophied morphology (enlarged cell body, reduced branching) and greater expression of M1 and M2 marker proteins [78,92,93,103]. This shift to a pro-inflammatory phenotype was accompanied by higher levels of pro-inflammatory cytokines—TNF-α, IL-1β, IL-6—and chemokines—CXCL10 and MCP-1 [52,78,90–92,107,108]. Additionally, the engagement of major immune signaling pathways was indicated by higher protein levels and phosphorylation of TLR4, ERK, NFκβ, and MYD88 in mice following microplastic exposure [78,92,93,108]. Studies have observed that microglia exert a strong functional response against microplastics in the brain, however this exposure eventually caused microglial cell death via pyroptosis and apoptosis [92,101]. Importantly, microglial activation and subsequent immune changes also occur secondary to microplastic-induced intestinal dysbiosis. As previously discussed, intestinal disruption causes peripheral inflammation and higher serum levels of LPS, both of which alter BBB permeability and induce neuroinflammation [29,100,109]. Within neuroimmune pathways, microglia have an important role in detecting and consuming microplastics in the brain as well as launching an inflammatory immune response when microplastics are present in the body. Neuroinflammation is a concerning side-effect of microplastic ingestion because prolonged neuroinflammation causes global changes to the structure and function of the brain.

### 5.3. Neurological Effects and Neurodegeneration

It has been shown that microplastic exposure is detrimental to neurons and thus affects brain function. For example, it has been reported that microplastics in vivo interfere with neurotransmission by altering the structure and function of synapses. In mice, microplastic exposure downregulates the expression of synaptogenic proteins and reduces the total number of synapses [98,101]. Similarly, microplastics significantly altered the expression

and activity of key neurotransmitters including dopamine, glutamate, serotonin, gamma-aminobutyric acid (GABA), and acetylcholine [9,93,101,102,110]. Further, microplastic exposure caused significant changes in enzymatic activity of acetylcholinesterase and acetylcholine transferase—both critical for cholinergic neurotransmission [98,101]. Despite this, locomotion changes have only been associated with microplastic exposure in nematodes [9,104,110], and not yet in mice [78]. Alarmingly, microplastics cause changes in brain tissues that are reflective of chronic degenerative brain pathologies, such as chronic traumatic encephalitis, PD, and AD [111–113]. On a gross level, the brain coefficient (brain-to-body mass ratio) of exposed mice has been negatively correlated with microplastic dose [91] and is evident in conditions including AD [114] and traumatic brain injury [115]. Additionally, oral microplastic exposure in chickens caused intra-cerebral haemorrhage within the granular layer, whereby the severity of haemorrhage was positively correlated with microplastic dose [108]. Structural changes have also been reflected in histological alterations including irregular cell arrangement, cytoplasmic vacuolisation, shrunken cell bodies, degeneration of dendritic spindles, and mitochondrial rupture in neuronal cell populations (Figure 2) [91,93,101,103]. This cellular damage in neuronal cells was associated with signs of oxidative stress, being the upregulation of reactive oxygen species (ROS) and ROS-related genes [9,93,101,102,113]. In a cortical spheroid model, six days of microplastic exposure did not significantly affect neuronal cells [102], however in other mouse and in vitro models, microplastics exposure (from 21 days in mice; 24 h in vitro) repeatedly induced cell death among neurons and glial cells [90,92,93,101,102]. Microplastic cytotoxicity in the brain of orally exposed mice was indicated by necrosis, more apoptotic cells, upregulated pro-apoptotic proteins (e.g., Bax and caspase 3), and downregulated anti-apoptotic proteins [91–93,103]. Among surviving cells, neurodegeneration resulted in the overall damage or loss of cholinergic neurons, GABAergic neurons [113], dopaminergic neurons, and glutamatergic neurons [9]. Widespread neuronal degeneration in exposed mice was also indicated by amplified pathways involved in neurodegenerative disease [103].

The implications of microplastics exposure in neurodegenerative disease have been investigated using in vivo models of AD and PD. In APP/PS1 double transgenic AD mice, microplastic exposure promoted neuroinflammation and microglial pyroptosis compared to unexposed AD mice [107]. Moreover, microplastic exposure in a nematode model of PD accelerated the degeneration dopaminergic neurons [104]. Recently, microplastics have also been found to induce α-synuclein aggregation, a key player in proteinopathies and neurodegenerative diseases. Microplastics interacted with α-synuclein directly in vitro, seeding the formation of fibrils and accelerating fibril growth [105]. Similarly, microplastics increased both the number of α-synuclein aggregates per cell and the total area of aggregates in dopaminergic neurons in a PD model [104]. Concerningly, microplastics that were co-injected with α-synuclein fibrils exhibited significantly greater dispersal in cultured neurons [105]. Additionally, nanoplastics increased the rate of nucleation among Amyloid β subtypes (a key AD hallmark) and promoted the formation of protein oligomers rather than fibrils, which are associated with neuronal membrane damage [116]. Taken together, these neurodegenerative disease models indicate that microplastics contribute to AD and PD pathology.

## 6. Current Methods and Limitations

Growing research using cell and animal models has begun to show that microplastics cause inflammation and dysfunction in the gut, brain, and around the body. However, the methodology adopted by these studies is diverse, and more realistic models of exposure are needed to gauge the true effect of microplastics on human health. Humans are exposed to microplastics for 73.4 years on average [117], but experimental exposure times range from 24 h [56,58,63,79,80] to four weeks [47,56,65,68,78,81], meaning that chronic or life-long exposure has not been well-replicated. Cell and animal studies have used a mix of pristine (primary) and artificially weathered microplastics—weathering techniques range from mechanical fragmentation and prolonged UV exposure to incubation in aquatic ecosystems

and digestive juices [9,49,69,118]. Fluorescent microplastics have been used to easily trace and quantify microplastic transport and deposition, but are not often manipulated to mimic environmental exposure or gastrointestinal digestion [58]—this would be a valuable tool for future studies. Overall, fragmented and weathered microplastics are more bioactive in cells and animals than the more commonly used spherical microplastics [48,57,81,119]; highlighting the need for further development and characterisation of environmentally realistic exposure models. Global demand for polyethylene, polypropylene, and polystyrene makes them the top three microplastic-forming polymers that contaminate oceans, drinking water, and staple food products [64,120]. However, most studies have used polystyrene microplastics because of their commercial availability, meaning that the health effects of other common polymers require further study. It is important to note that cell and tissues from in vitro, in vivo, and clinical studies contain a level of background microplastic contamination from laboratory equipment, such as polyvinyl chloride gloves and polypropylene pipette tips, or exposure in situ. This is not usually measured or considered, meaning most studies lack true negative controls. In cells and tissue samples from around the body, microplastics are quantified and analysed using techniques like confocal microscopy [31,56,58,59,96,104], Raman spectroscopy [8,10,54,87,98,121], and flow cytometry [96,102]. However, these tools are expensive and time consuming to use, require advanced skills, and are limited by the micrometre scale of microplastics—all of which are barriers to studying the effects of microplastics on health and disease. Although there is little standardisation across study methods, the field of microplastic-medical research is innovative and rapidly expanding.

## 7. Conclusions

Microplastic contamination in the environment is rising exponentially, as global plastic production continues to surge. Humans are exposed to more microplastics every year, and a growing variety of models have shown detrimental health effects, but the mechanisms of damage are still unclear. Nonetheless, the GBA is postulated as the link between microplastics and neurodegeneration. Orally ingested microplastics induce microbial dysbiosis and gut inflammation, resulting in peripheral inflammation and the circulation of PAMPs derived from intestinal microbes. Additionally, clinical and laboratory data show that microplastics accumulate in gut tissue and disperse into circulation, becoming embedded in tissues around the body and exacerbating a variety of chronic diseases. Microplastic-induced peripheral inflammation results in BBB disruption, facilitating microplastic dispersion into the brain, driving neurotoxicity and neurodegeneration. Functional neurological changes, neuroinflammation, and $\alpha$-synuclein protein pathology ensues, demonstrating that microplastic exposure could be an important contributor to the pathogenesis of neurodegenerative diseases like AD and PD. It seems that no system is spared from the consequences of microplastic exposure, yet there are still major limitations to studying the health effects of microplastics. Due to their size, even the most advanced technology is limited by time intensity, cost, and expertise. Additionally, it is difficult to mimic environmental weathering or chronic gastrointestinal exposure in relevant and reproducible ways. Hence, the development of more realistic exposure models that harness the traceability of commonly used fluorescent microplastics is a valuable direction for future studies. Understanding the effect of microplastic-induced gut disruption and the gut-brain-axis consequences of microplastic exposure is crucial to guide future research, therapeutic strategies, and policy changes aimed at mitigating the adverse consequences of these pervasive environmental contaminants.

**Author Contributions:** Conceptualization, C.E.S., A.M.G. and R.S.A.; Research, C.E.S.; Writing—Original Draft Preparation, C.E.S.; Writing—Review and Editing, C.E.S., A.M.G. and R.S.A.; Figure Production, C.E.S., A.M.G. and R.S.A.; Supervision, A.M.G. and R.S.A. All authors have read and agreed to the published version of the manuscript.

**Funding:** This research received no external funding.

**Institutional Review Board Statement:** Not applicable.

**Informed Consent Statement:** Not applicable.

**Data Availability Statement:** No new data was created or analyzed in this study. Data sharing is not applicable to this article.

**Acknowledgments:** Thanks to Chidozie Anyaegbu and Li Shan Chiu for their ongoing supervision. This research was supported by an Australian Government Research Training Program (RTP) Scholarship awarded to C. Sofield.

**Conflicts of Interest:** The authors declare no conflicts of interest.

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
