# Peer review of "Mind over Microplastics: Exploring Microplastic-Induced Gut Disruption and Gut-Brain-Axis Consequences"

_cimb, doi:10.3390/cimb46050256_

Round 1

Reviewer 1 Report

Comments and Suggestions for Authors

First of all, I would like to express my gratitude to CIMB journal and its editor for considering me as a reviewer for this paper. I find this article of great interest and relevance given its potential impact on public health policy. However, prior to publication, the authors should clarify some points:

- The document refers to "gut-first" alterations and "gut-first" neurodegenerative diseases on several occasions without explaining what it refers to. Please define this concept.

- In lines 106-108 it is stated that "Although microplastics are not toxic to the gut epithelium, the cellular stress that microplastic exposure induces has structural and functional consequences that threaten gut health overall."  The phrase appears contradictory. If exposure to microplastics generates cellular stress and production of ROS, it is unclear how it can be considered non-toxic to epithelial cells.

- Finally, in the limitations section, it may be worth mentioning that due to the widespread use of plastics in consumer goods, it is practically impossible to find individuals not exposed to microplastics in order to make comparisons. Additionally, it is important to note that even control cells in in vitro cultures are exposed to microplastics due to the use of disposable polycarbonate plates.

Author Response

Reviewer #1

First of all, I would like to express my gratitude to CIMB journal and its editor for considering me as a reviewer for this paper. I find this article of great interest and relevance given its potential impact on public health policy. However, prior to publication, the authors should clarify some points:

Response: We thank the reviewer for their constructive comments on this review.

  1. The document refers to "gut-first" alterations and "gut-first" neurodegenerative diseases on several occasions without explaining what it refers to. Please define this concept.

Response: Where appropriate, the term “gut first” has been re-phrased to avoid confusion.

  1. In lines 106-108 it is stated that "Although microplastics are not toxic to the gut epithelium, the cellular stress that microplastic exposure induces has structural and functional consequences that threaten gut health overall."  The phrase appears contradictory. If exposure to microplastics generates cellular stress and production of ROS, it is unclear how it can be considered non-toxic to epithelial cells.

Response: We agree with the reviewer’s comment and have rephrased this statement. The statement now reads:

Although microplastics are not lethal to cells of the gut epithelium, the cellular stress that microplastic exposure induces has structural and functional consequences that threaten gut health overall.”

  1. Finally, in the limitations section, it may be worth mentioning that due to the widespread use of plastics in consumer goods, it is practically impossible to find individuals not exposed to microplastics in order to make comparisons. Additionally, it is important to note that even control cells in in vitrocultures are exposed to microplastics due to the use of disposable polycarbonate plates.

Response:  We feel this is a valuable addition to the review, and greatly appreciate the reviewer’s suggestion. In the limitations paragraph the authors have incorporated discussion on the presence of background microplastic contamination in all common models of microplastic exposure.

“It is important to note that cell and tissues from in vitro, in vivo and clinical studies contain a level of background microplastic contamination from laboratory equipment, such as polyvinyl chloride gloves and polypropylene pipette tips, or exposure in situ. This is not usually measured or considered, meaning most studies lack true negative controls.”

Reviewer 2 Report

Comments and Suggestions for Authors

The detrimental effects of microplastics on health are long recognized, and dare I say – it’s a universal threat. Hence, microplastics are ubiquitous, and their impact on a gut’s homeostasis are described across species (GIT of finishes; Garcia-Torné et al 2023). Its long lifetime and low degradability, increasing plastic pollution, and climate change make a global concern. For this reason, I feel this is a timely and topical review.

1.      However, authors need to polish their writing skills. And I don’t mean “grammar and wording.” These are just fine. I literally mean the “skills of the writing.” Soften your style a bit and avoid the death giveaways of a “first-time writer.” - For this, I suggest most attentive proofreading. For instance, this manuscript has a ref#1 twice, with just one sentence between (lns 30 & 31).  What is the fundamental objective or methodology (we have been informed this is a review. What kind of review?)? For what it’s worth, this paper could just be a chaotic summary of everything.

2.      More importantly, this manuscript lacks any organizational logic—“mechanisms” should be before “effects”; events on the BBB could be easily described with the term “neuroinflammation,” possibly the missing link between the gut-brain axis and neurodegeneration.

3.      Above all,  strict adherence to the guidelines for authors is highly appreciated.  

Author Response

Reviewer #2

The detrimental effects of microplastics on health are long recognized, and dare I say – it’s a universal threat. Hence, microplastics are ubiquitous, and their impact on a gut’s homeostasis are described across species (GIT of finishes; Garcia-Torné et al 2023). Its long lifetime and low degradability, increasing plastic pollution, and climate change make a global concern. For this reason, I feel this is a timely and topical review.

Response: We thank the reviewer for their comments and agree with their assessment on the importance of such a review, acknowledging the increasing global impact of microplastics.

  1. However, authors need to polish their writing skills. And I don’t mean “grammar and wording.” These are just fine. I literally mean the “skills of the writing.” Soften your style a bit and avoid the death giveaways of a “first-time writer.” - For this, I suggest most attentive proofreading. For instance, this manuscript has a ref#1 twice, with just one sentence between (lns 30 & 31).  What is the fundamental objective or methodology (we have been informed this is a review. What kind of review?)? For what it’s worth, this paper could just be a chaotic summary of everything.

Response: As with all manuscripts submitted from the researchers, an extensive evaluation of the manuscript occurred at multiple stages. In relation to the first occurrence of reference 1, we have subsequently removed the first in-text citation (Line 27-28), as this is not required to justify the associated sentence. The authors have ensured extensive proofreading has been undertaken of this manuscript.

  1. More importantly, this manuscript lacks any organizational logic—“mechanisms” should be before “effects”; events on the BBB could be easily described with the term “neuroinflammation,” possibly the missing link between the gut-brain axis and neurodegeneration.

Response: Following the reviewer’s comment, we considered better organizing the manuscript’s narrative, and altered the paragraph order. We agree that one logical approach would be to discuss mechanisms prior to effects. However, the evidence for mechanisms of microplastic uptake and damage is scarce and largely inconclusive at present. For this reason, we have focussed on the effects of microplastic exposure and expanded on possible mechanisms where reasonable. We feel that “events on the BBB” (structural and functional changes resulting in altered permeability) are distinct from neuroinflammation–which has been comprehensively described with a particular focus on microglial activation. We agree that BBB changes are pivotal in linking the GBA and neurological damage, and this has informed the structure of our discussion.

Examples of major revisions to structure include but are not limited to:

  • New order of information in the section ‘The Effects of Microplastics in Gut Health’, which follows the effects of microplastics from the gut lumen to the epithelium and mucosa and now reads: Introducing the Gut-Brain-Axis > Mechanisms of Microplastic Damage to the Gut > The Gut Microbiome > The Gut Microbiome: Function Effects > Uptake of Microplastics from the Gut Lumen > Mechanisms of Microplastic Uptake > Environmental and Luminal Factors Influence Microplastic Accumulation in the Gut > Microplastic Toxicity > Histological Alterations in the Gut > Enteric Immune Activation
  • The authors have substantially re-organised the entire paragraph (413-445) and have emphasised the transition into the discussion of microplastic exposure in disease models specifically.
  1. Above all, strict adherence to the guidelines for authors is highly appreciated.  

Response: The authors followed the Journal requirements and referencing style and appreciate the positive feedback for this.

Reviewer 3 Report

Comments and Suggestions for Authors

General comment:

Overall the review is well written and interesting, the work done is supported by solid and numerous references. The advice I want to give to the authors is to better organize the information reported, avoiding excessive repetition of concepts or confusion.

in particular:

line 81-82 and 130-132: explain or at least anticipate the link between leaky gut and interference with BBB homeostasis and neurological complications.

line 180-181: The reduction of bacterial types is linked to the increase of the inflammatory state (intervention of immune cells, environment hostile to bacterial proliferation, etc.) or are there any evidence that the effect of microplastics on the characteristics of viability and bacterial populations?

line 242-248: the description of the different effect of the microplastics according to size and shape is reported several times, therefore I recommend to place this description in the initial part or crumple in this part, avoiding excessive repetition.

line 340-346: As above, integrate this information without repeating it in the manuscript. I also suggest reorganising the information so as to report the studies without switching from live to in vitro and vice versa, one must organize well the many observations reported to avoid confusion.

line 351-353: describe this part more extensively, very interesting in the context of this review.

Neurodegeneratrion: in this paragraph, really very interesting is essential to better organize information and reduce repetitions.

Author Response

 Reviewer #3

Overall the review is well written and interesting, the work done is supported by solid and numerous references. The advice I want to give to the authors is to better organize the information reported, avoiding excessive repetition of concepts or confusion.

Response: We thank the reviewer for their insightful feedback on this review. We have critically considered the organization of information and have made major revisions throughout the review to reduce repetitions.

  1. line 81-82 and 130-132: explain or at least anticipate the link between leaky gut and interference with BBB homeostasis and neurological complications.

Response: The authors have introduced the link between leaky gut, GBA disruption and BBB permeabilization (line 89 and line 262-264), although we acknowledge that this is brief - to avoid repeating information that is described in more detail later in the review.

  1. line 180-181: The reduction of bacterial types is linked to the increase of the inflammatory state (intervention of immune cells, environment hostile to bacterial proliferation, etc.) or are there any evidence that the effect of microplastics on the characteristics of viability and bacterial populations?

Response: The reviewer raises an interesting point which we have now expanded on, to address the comment.

Line 99-105 has described the direct effect of microplastics on bacterial growth and community structure in environmental settings. The mechanism of microplastic-dysbiosis in the gut is not clear in the literature, however evidence of microplastic interactions with the gastric pathogen Helicobacter pylori has been discussed (line 216-219).

  1. line 242-248: the description of the different effect of the microplastics according to size and shape is reported several times, therefore I recommend to place this description in the initial part or crumple in this part, avoiding excessive repetition.

Response: The effects of size and shape (line 167-182 ) has been re-ordered and condensed as suggested by the reviewer.

  1. line 340-346: As above, integrate this information without repeating it in the manuscript. I also suggest reorganising the information so as to report the studies without switching from live to in vitro and vice versa, one must organize well the many observations reported to avoid confusion.

Response: The authors have condensed and re-ordered the information (line 342-349) to improve flow and reduce repetition. However, we have been mindful of maintaining adequate detail in each section for the benefit of readers, given the scope of this review.

  1. line 351-353: describe this part more extensively, very interesting in the context of this review.

Response: The information has been re-ordered to emphasise the role of microplastic-induced metabolic alterations in peripheral disease and GBA dysfunction (line 349-358). We felt that expanding on this further was outside the scope of this review, although this is an intriguing side-effect of microplastic exposure.

  1. Neurodegeneratrion: in this paragraph, really very interesting is essential to better organize information and reduce repetitions.

Response: The authors have substantially re-organised the entire paragraph (413-445) and have emphasised the transition into the discussion of microplastic exposure in disease models specifically.

Round 2

Reviewer 2 Report

Comments and Suggestions for Authors

I appreciate the invitation!